# Direction of Arrival Method for L-Shaped Array with RF Switch: An Embedded Implementation Perspective [note 1]

**DOI:** 10.3390/s23063356

**Published:** 2023-03-22

**Authors:** Tiago Troccoli, Juho Pirskanen, Jari Nurmi, Aleksandr Ometov, Jorge Morte, Elena Simona Lohan, Ville Kaseva

**Affiliations:** 1Faculty of Information Technology and Communication Sciences, Tampere University, 33720 Tampere, Finland; 2WIREPAS Ltd., 33720 Tampere, Finland

**Keywords:** Direction of Arrival (DOA), Multiple Signal Classification (MUSIC), Internet of Things (IoT), embedded systems, array signal processing

## Abstract

This paper addresses the challenge of implementing Direction of Arrival (DOA) methods for indoor localization using Internet of Things (IoT) devices, particularly with the recent direction-finding capability of Bluetooth. DOA methods are complex numerical methods that require significant computational resources and can quickly deplete the batteries of small embedded systems typically found in IoT networks. To address this challenge, the paper presents a novel Unitary R-D Root MUSIC for L-shaped arrays that is tailor-made for such devices utilizing a switching protocol defined by Bluetooth. The solution exploits the radio communication system design to speed up execution, and its root-finding method circumvents complex arithmetic despite being used for complex polynomials. The paper carries out experiments on energy consumption, memory footprint, accuracy, and execution time in a commercial constrained embedded IoT device series without operating systems and software layers to prove the viability of the implemented solution. The results demonstrate that the solution achieves good accuracy and attains an execution time of a few milliseconds, making it a viable solution for DOA implementation in IoT devices.

## 1. Introduction

Direction of Arrival (DOA) estimation is an array signal processing application found in radars, navigation, military devices, and medical appliances [1]. Most recently, it has been applied in wireless communication systems such as Bluetooth to localize devices in indoor environments where satellite-based radio-navigation systems fail to do so. In 2015, Bluetooth released its positioning technology based on the Received Signal Strength (RSS) [2]. In that past technology, a Bluetooth Low Energy (BLE) tag transmitted radio frequency (RF) signals to receivers. The receivers could estimate the positions of such tags from RSS measurements. The positions were not precisely estimated, in fact, the tag could be anywhere inside a circular zone if trilateration was used, which is a typical RSS-based positioning method. According to Bluetooth itself, such technology has an accuracy of a few meters (1–10 m) [3], and even if more complex algorithms are employed, there are limitations on the achievable positioning accuracy. In some cases, such accuracy of a few meters is enough, but some occasions need higher accuracy, for example, machine navigation for autonomous mobile robots, drones, industrial automation, or navigation systems that guide people in indoor environments. Higher accuracy is also desirable in asset tracking, where factories track material workflow and hospitals track equipment location.

Most recently, Bluetooth has released a new capability that makes it possible to estimate DOAs [4], unlocking high accuracy as DOA-based positioning is reported to attain sub-meter-level position accuracy [3,5,6,7]. In the direction of arrival version of that new capability, receivers have an array of antennas to make it possible to estimate the azimuth and elevation angles coming from a signal emitted from a BLE tag. By applying, for example, the triangulation method, which employs the cited angles for each receiver and their positions, it is possible to estimate the location of BLE tags. Besides using for DOA estimation, BLE is broadly used for wireless communication of the Internet of Things (IoT). In the IoT terminology, its networks are composed of many nodes that are small battery-powered resource-constrained embedded systems. Particularly, in IoT networks that deploy DOA-based positioning systems, some nodes, called anchor nodes, have an array of antennas that execute DOA methods to estimate the azimuth and elevation of RF signals emitted from BLE tags which are also nodes. Figure 1 depicts an IoT mesh network with the cited positioning systems.

In IoT networks, it is important to note that nodes are often constrained embedded systems. These systems typically consist of three distinct subsystems [8], as illustrated in Figure 2. The microcontroller or computing subsystem is responsible for controlling the node’s functionalities, and runs instructions on a low-power processor that commonly operates at a few megahertz. It also has flash memory and RAM whose sizes are in order of kilobytes. Flash memory is a non-volatile memory that stores the program’s instructions and constant data values, whereas RAM is a volatile memory that is used as data storage for the program while it is running. Nodes usually have a simple real-time operating system that executes all functionalities, also known as tasks, concurrently. The sensor subsystem collects data from natural phenomena in the form of analog signals via sensor readings and transforms them into digital signals using an Analog-to-Digital Converter (ADC). Finally, the communication subsystem is made up of a transceiver and normally a co-processor device that is responsible for transmitting and receiving data.

While Bluetooth specifies a protocol for transmitting a constant tone for DOA purposes, the development of direction of arrival algorithms is left to implementation. However, the implementation of DOA methods in IoT devices presents a significant challenge as these devices are often battery-powered and resource-constrained embedded systems with limited computational resources as previously mentioned. In contrast, DOA methods consist of complex numerical algorithms that are resource-intensive and time-consuming, potentially leading to rapid battery depletion, unacceptable execution times, and resource starvation. Moreover, IoT devices typically run a simple real-time operating system that executes small tasks concurrently, such as data acquisition from sensors and communication with other devices. The execution of DOA methods in such a multi-threaded environment is even more challenging, and computational resource management needs to be carefully thought out.Therefore, developing DOA algorithms for IoT devices demands an innovative approach that considers the balance between resource constraints and DOA accuracy, without compromising battery life or real-time system performance.

## 2. Literature Review

Typically, research about DOA does not take into account the implementation point of view sufficiently. In many cases, research is carried out using multi-paradigm programming languages such as MATLAB, which already have a range of pre-made general-purpose numerical functions, resulting in little motivation to develop tailor-made numerical algorithms for DOA methods. However, when implementing these methods in constrained embedded systems, numerical algorithms must be developed from scratch, as C programming language libraries offer very limited support. Additionally, widely-used linear algebra libraries such as LAPACK [9] and Armadillo [10] are not suitable for use in constrained embedded systems. Although the Common Microcontroller Software Interface Standard (CMSIS) DSP Software Library is designed for these systems, it lacks some numerical algorithms used in DOA methods. Furthermore, DOA estimations are typically used in radars or large antenna arrays that employ much more powerful processors than those in constrained embedded systems. While accuracy is usually the primary performance criterion of interest, energy consumption, and memory usage are also crucial considerations for battery-powered IoT devices. Therefore, DOA methods should not occupy significant amounts of memory, blocking other IoT tasks. Additionally, complex numerical algorithms require considerable computation, which affects the execution time, an essential performance criterion for real-time applications.

The array of antenna used in this research is the L-shaped array. The L-shaped array has an interesting propriety since it is composed of two orthogonal Uniform Linear Arrays (ULAs). One-dimensional  (1D) DOA methods can be applied separately for two ULAs to estimate the azimuth and elevation angles. Other shapes of planar arrays, such as Uniform Rectangular Arrays (URAs) and Uniform Rectangular Frame Arrays (URFAs), rely on two-dimensional (2D) DOA methods which are more complicated than their 1D versions. Moreover, well-known fast algorithms were specifically developed for ULAs. Additionally, some exploit the Vandermonde structure in the signal model found in such an array to speed up their computations. Namely, Root Multiple Signal Classification (Root MUSIC) [11], Estimation of Signal Parameters via Rotational Invariant Techniques (ESPRIT) [12], Fourier Domain MUSIC [13], and Rank-Reduction Method (RARE) [14]. Most recently, many modified versions of the cited methods have been devised, claiming to be a better version in a certain way. Nevertheless, ESPRIT was later designed for URAs and Uniform Circular Arrays (UCAs) [15], whereas Fourier Domain MUSIC can be applied to non-uniform linear arrays as well.

Among many DOA methods, Multiple Signal Classification (MUSIC) [16], invented in the 80s, is an important algorithm that has been extensively tested in simulation and the real world for many decades as well as comprehensively studied, culminating in some well-known modified versions. Notably, the standard MUSIC detects DOAs by searching for peaks in the spatial spectrum. It also can be extended to find azimuth and elevation angles by searching for peaks in a 2D spatial spectrum of the planar array of antennas. However, that 2D search is a prohibitively expensive operation which motivated the development of Reduced-Dimension (R-D) MUSIC [17] that exploits the structure of an L-shaped array of antennas to do that search in two 1D spatial spectra, one for each ULA. Knowing Root-MUSIC is a search-free method that exploits the Vandermonde structure of ULAs to apply a root-finding method that substantially reduces its execution time, the Reduced-Dimension (R-D) Root MUSIC came naturally as a modified version that executes Root MUSIC two times, one execution for each ULA.

Research about the BLE direction finding is still scarce as it is a recent feature released in 2017. In [18], research was conducted involving real-world experiments using that BLE feature in an indoor environment focused on array calibrations using two ULAs and one URA. It reported that mutual coupling had a minimal influence or even had no clear improvement on the accuracy of a variation of MUSIC. However, Carrier Frequency Offset (CFO) compensation substantially impacted it. In [5], four 4 × 4 URAs were employed in an indoor environment of 25 m × 15 m wide composed of pillars, walls, human movements, tables, chairs, devices, and lamps. In total, 8 eight distinct positions were estimated, and 32 different estimations were evaluated using BLE direction finding and the MUSIC method. The average error was 0.7 m, attaining a good result concerning the sub-1-meter accuracy purpose.

Papers about real-world implementations are less common than about in a simulation. In [19], a modified version of MUSIC was developed in a Digital Signal Processor (DSP) for underwater acoustic sources. Notably, it employed a Reduced-Order Root-MUSIC method that reduces the polynomial order of Root MUSIC to speed up computations. In [20,21], researchers conducted a small-scale experiment using a ULA with four elements and one single transmitter. The array of antennas was connected to the NI PXI platform, and an antenna transmitter was connected to another PX platform. After running a single-source MUSIC and a Total Least Squares ESPRIT in LabView NI hardware to estimate two different DOAs, they concluded both methods could be used in real-time applications. Moreover, the development of 2D MUSIC for an L-shaped antenna array to estimate the azimuth and elevation angles based on parallel computing was successfully devised for a Digital Signal Processor (DSP) [22]. More specifically, researchers parallelized the eigenvalue decomposition to construct the signal or noise subspace and the peak searching method to find DOAs by exhaustive search. Despite the parallelization, the execution time of parallel MUSIC was 190.39 ms, which can be seen as slow considering that the experiment used a DSP of 1 GHz.

Most of the research did not focus on the implementation aspect regarding the optimization of the method to be adapted to constrained embedded systems. Except for one cited paper, the rest did not evaluate other important performance criteria such as execution time, energy consumption, and memory footprint. This research takes a step further by adapting the R-D Root MUSIC for a Radio Frequency (RF) switch based on Bluetooth specification. Additionally, the adapted method applies unitary transformations to avoid complex arithmetic, and a real-valued Eigenvalue Decomposition Method (EVD) is employed to find the roots of complex-valued polynomials. The novel method also exploits the radio communication system design of Bluetooth to speed up its execution based on BLE receivers’ capability of estimating one single DOA only.

**Notation:** by defining the complex number $z=ce^{j\theta }$, the operators Re(·) and Im(·) return the real and imaginary part of *z*, respectively. The argument of a complex number arg(·) is an operator that returns the phase angle (θ) in the interval [−π,π] while $\bar{\left(\cdot \right)}$ is the complex conjugate. The complex absolute operator is defined as |·|=Re(·)2+Im(·)2. The operator diag(·) denotes a diagonal matrix formed by an input vector. The Hermitian transpose is ${\left(\cdot \right)}^{H}$ which applies the transpose and complex conjugate on a matrix. The operator ⌊·⌋ is the floor function, that is, it outputs the greatest integer less than or equal to the input which is a real number. The Hadamard product (∘) is an operator that takes two matrices, for example, A and B of the same dimension, and outputs another matrix whose elements are given by ${\left(A\circ B\right)}_{ij}=\left(A_{ij}\right)\left(B_{ij}\right)$. $I_{n}$ is an identity matrix while $0_{n}$ is a zero column vector both of size *n*.

## 3. Mathematical Model for L-Shaped Uniform Array

Figure 3 shows the structure of the L-shaped uniform array. It is composed of two orthogonal uniform linear arrays of *M* antennas in the X-Y plane in which the distance between two adjacent antennas is Δ. All antennas are assumed to be identical, isotropic, and omnidirectional. Suppose there are *d* (d<M) independent far-field narrowband stationary signals, $s_{i}\left(t\right)$ such that i=1,⋯,d, incident on the array plane at 2D angle $\left(\theta_{1},\phi_{1}\right),\left(\theta_{2},\phi_{2}\right),\cdots ,\left(\theta_{d},\phi_{d}\right)$ in which $\theta_{i}$ is the azimuth and $\phi_{i}$ is the elevation angle. Let us also assume the signals propagate in an AWGN channel with linear and isotropic transmission medium. DOA methods compute the broadside angle, which is the signal direction measured relative to the line perpendicular to the array. Since that array is composed of two ULAs, there are two broadside angles, $\alpha_{i}$, and $\beta_{i}$, as shown in Figure 3, which correspond to the x-axis and y-axis ULAs, respectively. From geometric properties, the relation between $\alpha_{i}$ and $\beta_{i}$ with azimuth and elevation are shown in Equations (Equation 1) [23],
(1)$$\begin{array}{cc} \cos\alpha_{i} & =\cos\theta_{i}\sin\phi_{i},\\ \cos\beta_{i} & =\sin\theta_{i}\sin\phi_{i}. \end{array}$$

Assume that the L-shaped array is not subject to nonidealities such as mutual coupling and cross-polarization effect. Additionally, consider that all antennas are identical and have omnidirectional gain functions, i.e., g(θ,ϕ)=1. The model of the signal samples received by x-axis and y-axis arrays at a timestamp *t* can be expressed as Equations (Equation 2) and (Equation 3) [24],
(2)$$x\left(t\right)=A_{x}s\left(t\right)+n_{x}\left(t\right),$$
(3)$$y\left(t\right)=A_{y}s\left(t\right)+n_{y}\left(t\right),$$
where $x\left(t\right)={\left[\begin{array}{cccc} x_{1}\left(t\right) & x_{2}\left(t\right) & \cdots & x_{M}\left(t\right) \end{array}\right]}^{T}$ is the array observation at timestamp *t* of the x-axis ULA which is a vector of the signal samples for each individual antenna in the x-axis ULA, such that $x_{i}\left(t\right)$ corresponds to a single signal sample received from the antenna *i* at timestamp *t*. Likewise, for $y\left(t\right)={\left[\begin{array}{cccc} y_{1}\left(t\right) & y_{2}\left(t\right) & \cdots & y_{M}\left(t\right) \end{array}\right]}^{T}$ in the case of y-axis ULA. Moreover, $s\left(t\right)\in C^{d\times 1}$ is a vector of signals of *d* sources, $n_{x}\left(t\right),n_{y}\left(t\right)\in C^{M\times 1}$ are the additive white Gaussian noise (AWGN) and $A_{x},A_{y}\in C^{M\times d}$ are the ideal steering matrices of the x-axis array and y-axis array, respectively, as shown in Equations (Equation 4) and (Equation 5),
(4)$$A_{x}=\left[\begin{array}{cccc} a(\alpha_{1}) & a(\alpha_{2}) & \cdots & a(\alpha_{d}) \end{array}\right]$$
(5)$$A_{y}=\left[\begin{array}{cccc} a(\beta_{1}) & a(\beta_{2}) & \cdots & a(\beta_{d}) \end{array}\right]$$
where the ideal array responses are defined in Equations (Equation 6) and (Equation 7),
(6)$$a{\left(\alpha_{i}\right)}^{T}=\left[\begin{array}{ccccc} 1 & e^{j\mu_{\alpha_{i}}} & e^{j2\mu_{\alpha_{i}}} & \cdots & e^{j\left(M-1\right)\mu_{\alpha_{i}}} \end{array}\right],$$
(7)$$a{\left(\beta_{i}\right)}^{T}=\left[\begin{array}{ccccc} 1 & e^{j\mu_{\beta_{i}}} & e^{j2\mu_{\beta_{i}}} & \cdots & e^{j\left(M-1\right)\mu_{\beta_{i}}} \end{array}\right],$$
in which $\mu_{\alpha_{i}}=-\frac{2\pi f_{c}}{c}\Delta \cos\theta_{i}\sin\phi_{i}=-\frac{2\pi f_{c}}{c}\Delta \cos\alpha_{i}$ and $\mu_{\beta_{i}}=-\frac{2\pi f_{c}}{c}\Delta \sin\theta_{i}\sin\phi_{i}=-\frac{2\pi f_{c}}{c}\Delta \cos\beta_{i}$, and *c* is the speed of light, $f_{c}$ is the carrier frequency.

Moreover, in our analyses, it would be useful in some cases to consider the model signal samples for the whole L-shaped array instead of two separate ULAs. Let us define $h\left(t\right)={\left[\begin{array}{cccc} h_{1}\left(t\right) & h_{2}\left(t\right) & \cdots & h_{2M-1}\left(t\right) \end{array}\right]}^{T}$ as the array observation of the L-shaped array at timestamp *t* which is a vector of the signal samples for each individual antenna in the L-shaped array, such that $h_{i}\left(t\right)$ corresponds a single signal sample received from the antenna *i* at timestamp *t*. The signal samples of an L-shaped array are composed of $h_{i}\left(t\right)=x_{i}\left(t\right)$ such that i=1,⋯,M−1, the common antenna $h_{M}\left(t\right)=x_{M}\left(t\right)=y_{1}\left(t\right)$ for both ULAs in addition to $h_{M+j-1}\left(t\right)=y_{j}\left(t\right)$ such that j=2,⋯,M. The model of the signal samples received by the L-shaped array at a timestamp *t* can be expressed as Equation (Equation 8),
(8)$$h\left(t\right)=A_{h}s\left(t\right)+n_{h}\left(t\right),$$
where $n_{h}\left(t\right)\in C^{(2M-1)\times 1}$ is the additive white Gaussian noise (AWGN) and $A_{h}\in C^{(2M-1)\times d}$ is the ideal steering matrix of the L-shaped array, respectively, shown in Equation (Equation 9),
(9)$$A_{h}=\left[\begin{array}{cccc} a(\alpha_{1},\beta_{1}) & a(\alpha_{2},\beta_{2}) & \cdots & a(\alpha_{d},\beta_{d}) \end{array}\right],$$
if each element of the ideal L-shaped array response, $a\left(\alpha ,\beta \right)\in C^{(2M-1)\times 1}$, is represented in Equation (Equation 10),
(10)$$a_{k_{x},k_{y}}=e^{j\left(k_{x}-1\right)\mu_{\alpha }}e^{j\left(k_{y}-1\right)\mu_{\beta }},$$
where $1\leq k_{x}\leq M$ and $1\leq k_{y}\leq M$, hence the L-shaped array response can be expressed in a compact form in accordance with Equation (Equation 11),
(11)$$a{\left(\alpha ,\beta \right)}^{T}={\left[\begin{array}{cccccccc} a_{1,1} & a_{2,1} & \cdots & a_{M,1} & a_{M,2} & a_{M,3} & \cdots & a_{M,M} \end{array}\right]}^{T}.$$

## 4. Unitary Reduced-Dimension Root MUSIC

In the standard 2D MUSIC, the array observation of both x-axis and y-axis ULAs are combined in such a way that it is possible to estimate the azimuth and elevation angle of each signal source by performing a 2D search on the spatial spectrum. In contrast, the Unitary Reduced-Dimension (R-D) Root MUSIC computes the two ULAs separately by applying a root-finding method that does not require a 2D search on the spatial spectrum. Consequently, that method is substantially faster than standard 2D MUSIC [17]. Moreover, the Unitary transformation turns centro-hermitian matrices into real-valued ones which further decreases the execution time and reduces memory consumption.

For the standard 2D MUSIC, let z(t) be the combination of the array observation of the x-axis and y-axis in a timestamp *t* as defined in Equation (Equation 12),
(12)$$z\left(t\right)=\left[\begin{array}{c} x(t)\\ y(t) \end{array}\right].$$

After collecting *N* array observations, the method computes its sample covariance matrix expressed in Equation (Equation 13),
(13)$$R_{zz}\approx {\hat{R}}_{zz}=\left(\frac{1}{N}\right)ZZ^{H},$$
such that $Z=\left[\begin{array}{cccc} z(t_{1}) & z(t_{2}) & \cdots & z(t_{N}) \end{array}\right]\in C^{2M\times N}$. Under some assumptions [24], the covariance matrix, $R_{zz}$, is non-singular and its eigenvectors are divided into two subspaces: a noise subspace ($U_{N}$) and a signal subspace ($U_{S}$). More specifically, the signal subspace ($U_{S}$) is composed of eigenvectors corresponding to the *d* largest eigenvalues. While the noise subspace ($U_{N}$) is composed of eigenvectors corresponding to the N−d smallest eigenvalues. Ideally, the N−d smallest eigenvalues are zeros; however, due to AGWN, they are nonzeros. Moreover, the noise subspace is orthogonal to the combined steering vector as indicated by Equation (Equation 14),
(14)$$U_{N}^{H}\left[\begin{array}{c} a(\alpha_{i})\\ a(\beta_{i}) \end{array}\right]=0,\;i=1,\cdots ,d.$$

From Equation (Equation 14), the spatial spectrum of 2D MUSIC method can be denoted in Function (Equation 15),
(15)$$P_{2D-MUSIC}\left(\alpha ,\beta \right)=\frac{1}{{\left[\begin{array}{c} a(\alpha )\\ a(\beta ) \end{array}\right]}^{H}U_{N}U_{N}^{H}\left[\begin{array}{c} a(\alpha )\\ a(\beta ) \end{array}\right]}$$
and the angles $(\alpha_{i},\beta_{i}),\;i=1,\cdots ,d$ can be found by performing a 2D search of the *d* peaks on Equation (Equation 15). As previously mentioned, since the 2D search peak finding is prohibitively time consuming, the Unitary R-D Root MUSIC overcomes this problem by performing a root-finding method instead. Before describing the Unitary R-D Root MUSIC, we have to define three matrices that are used in the unitary transformation to convert centro-hermitian matrices into real-valued ones. Let $\Pi_{p}\in C^{p\times p}$ be any anti-diagonal identity matrix in keeping with Definition (Equation 16),
(16)$$\Pi_{p}≜\left[\begin{array}{ccccc} 0 & 0 & \cdots & 0 & 1\\ 0 & 0 & \cdots & 1 & 0\\ \cdots & \cdots & \cdots & \cdots & \cdots \\ 0 & 1 & \cdots & 0 & 0\\ 1 & 0 & \cdots & 0 & 0 \end{array}\right],$$
and $Q_{n}\in C^{n\times n}$ be an unitary transform matrix expressed in Definitions (Equation 17) and (Equation 18),
(17)$$Q_{2n}≜\frac{1}{\sqrt{2}}=\left[\begin{array}{cc} I_{n} & jI_{n}\\ \Pi_{n} & -j\Pi_{n} \end{array}\right]\;\mathrm{or}$$
(18)$$Q_{2n+1}≜\frac{1}{\sqrt{2}}=\left[\begin{array}{ccc} I_{n} & 0 & jI_{n}\\ 0_{n}^{T} & \sqrt{2} & 0_{n}^{T}\\ \Pi_{n} & 0 & -j\Pi_{n} \end{array}\right],$$
depending if its size is even or odd. The algorithm is outlined below.

1.Collect *N* array observations for timestamp $t_{1},t_{2},\cdots ,t_{N}$. Afterward, we can estimate the covariance matrix of the x-axis and y-axis separately in line with Equations (Equation 19) and (Equation 20),
(19)$$R_{xx}\approx {\hat{R}}_{xx}=\left(\frac{1}{N}\right)XX^{H},$$
(20)$$R_{yy}\approx {\hat{R}}_{yy}=\left(\frac{1}{N}\right)YY^{H},$$
where $X=\left[\begin{array}{ccc} x(t_{1}) & \cdots & x(t_{N}) \end{array}\right]\in C^{M\times N}$ and $Y=\left[\begin{array}{ccc} y(t_{1}) & \cdots & y(t_{N}) \end{array}\right]\in C^{M\times N}$ are the *N* array observations of x-axis and y-axis respectfully.2.Apply forward-backward averaging on the covariance matrix, that is, ${\hat{R}}_{xx}^{fb}={\hat{R}}_{xx}+\Pi_{M}\bar{{\hat{R}}_{xx}}\Pi_{M}$ to deal with coherent signals due to the multipath reflections [25]. Since ${\hat{R}}_{xx}^{fb}$ is a centro-hermitian matrix, we apply the unitary transformation to convert that complex-valued matrix into a real-valued one, that is, C^x={QMHR^xxfbQM}=Re{QMHR^xxQM}∈RM×M. The same goes for the covariance matrix of the y-axis, C^y=Re{QMHR^yyQM}∈RM×M.3.Apply the real-valued EVD in $C_{x}$ and $C_{y}$ to construct the noise subspace $U_{N,x}$ and $U_{N,y}$ of the x-axis and y-axis, respectively. It is a time-consuming operation that is optimized in this paper.4.As previously mentioned, the Unitary R-D Root MUSIC estimates the DOAs based on the roots of two polynomials, so it avoids the exhaustive search of standard MUSIC. Defining $z=e^{-j\frac{2\pi }{\lambda }\Delta \cos\alpha }$ for the x-axis, the array response in Equation (Equation 6) can be redefined as a function of *z*, as shown in Equation (Equation 21),
(21)$$a{\left(\alpha \right)}^{T}=a{\left(z\right)}^{T}=\left[\begin{array}{ccccc} 1 & z & z^{2} & \cdots & z^{M-1} \end{array}\right],$$
likewise for the y-axis if we define $z=e^{-j\frac{2\pi }{\lambda }\Delta \cos\beta }$. In addition, since $a{\left(z\right)}^{H}=a^{T}\left(\frac{1}{z}\right)$, the MUSIC spectrum can be viewed as a polynomial function whose DOA information is contained in some of its roots. The polynomial of the Unitary R-D Root MUSIC for the x-axis and y-axis are defined in Equations (Equation 22) and (Equation 23) [11],
(22)$$p_{x}\left(z\right)=z^{M-1}a^{T}\left(\frac{1}{z}\right)Q_{M}U_{N,x}{\left(U_{N,x}\right)}^{H}Q_{M}^{H}a\left(z\right),$$
(23)$$p_{y}\left(z\right)=z^{M-1}a^{T}\left(\frac{1}{z}\right)Q_{M}U_{N,y}^{y}{\left(U_{N,y}\right)}^{H}Q_{M}^{H}a\left(z\right),$$
whose degree is 2M−2. By defining $G=Q_{M}U_{N,x}{\left(U_{N,x}\right)}^{H}Q_{M}^{H}$, we can derive Equations (Equation 24),
(24)$$\begin{array}{cc} p_{x}\left(z\right) & =z^{M-1}\left[\begin{array}{ccccc} 1 & z^{-1} & z^{-2} & \cdots & z^{-(M-1)} \end{array}\right]G{\left[\begin{array}{ccccc} 1 & z & z^{2} & \cdots & z^{M-1} \end{array}\right]}^{T}\\ \; & =\left[\begin{array}{ccccc} z^{M-1} & z^{M-2} & z^{M-3} & \cdots & 1 \end{array}\right]\left[\begin{array}{ccc} g_{1,1} & \cdots & g_{1,M}\\ \vdots & \cdots & \vdots \\ g_{M,1} & \cdots & g_{M,M} \end{array}\right]\left[\begin{array}{c} 1\\ z\\ z^{2}\\ \vdots \\ z^{M-1} \end{array}\right]\\ \; & =a_{0}+a_{1}z+\cdots +a_{2M-2}z^{2M-2}, \end{array}$$
whose coefficients are defined in the Relation (Equation 25),
(25)$$a_{k-1}=\left\{\begin{array}{cc} \sum_{i=1}^{k}g_{i,M-k+i} & \mathrm{if}\;k=1,2,\cdots ,M\\ \sum_{i=1}^{(2M-1)-k+1}g_{i,M-k+i} & \mathrm{if}\;k=M+1,M+2,\cdots ,2M-1. \end{array}\right.$$The same procedure goes to $p_{y}\left(z\right)$.5.The *d* complex roots of $p_{x}\left(z\right)$ and $p_{y}\left(z\right)$ that are inside of a unit circle and closest to it, namely, ${\hat{z}}_{x,1},{\hat{z}}_{x,2},\cdots ,{\hat{z}}_{x,d}$ and ${\hat{z}}_{y,1},{\hat{z}}_{y,2},\cdots ,{\hat{z}}_{y,d}$, respectively, contain information about the *d* DOAs. The azimuth and elevation angles can be found as indicated by the set of equation in (Equation 26),
(26)$$\begin{array}{cc} {\hat{\theta }}_{i} & =\arctan\left(\frac{\arg({\hat{z}}_{y,i})}{\arg({\hat{z}}_{x,i})}\right),\\ {\hat{\phi }}_{i} & =\arcsin\left(\frac{\lambda }{2\pi \Delta }\sqrt{(\arg{\left({\hat{z}}_{x,i}\right)}^{2}+\arg{\left({\hat{z}}_{y,i}\right)}^{2}}\right),\\ \forall i & =1,2,\cdots ,d. \end{array}$$Note that finding all roots of a complex-valued polynomial is a difficult task and time-consuming, thus it needs an efficient and accurate root-finding method. The implemented solution circumvents complex arithmetic and finds them by a real-valued EVD, which is described in Section 6.

## 5. RF Switch Model

Theoretically, all antennas within an array should sample the signal at the same time at each antenna port. However, this would require each antenna to have its own RF front-end, which includes components such as analog-to-digital converters, filters, mixers, and low-noise amplifiers. Unfortunately, incorporating such analog components for each antenna would lead to increased power consumption, physical size, and higher costs for constrained embedded IoT devices. To address these challenges, it is more appropriate for the array to have a single RF front-end and an RF switch, enabling each antenna to utilize the RF front-end at different times. The Bluetooth protocol considers this radio architecture for its direction-finding capability, and in this research, we opted to utilize the switching protocol outlined in the Bluetooth 5.1 specification [4] with an L-shaped array.

Bluetooth utilizes Gaussian Frequency-Shift Keying (GFSK) where 0s and 1s are modulated into different frequency shifts [26]. The two frequencies are equal to the central frequency ($f_{c}$) in addition to a frequency deviation ($\pm f_{\Delta }$). To comply with the theoretical assumption, the signal should be stationary, that is, a signal with a constant time-frequency is desirable. Thus, the transmitter (signal source) sends a Constant Tone Extension (CTE), composed of a continuous series of digital ones, so the frequency remains the same during the IQ sampling. Notably, if the signal were non-stationary, DOA methods would be more complex [27].

During the CTE, the first 4 μs is the guard period. The reference period, which takes 8 μs, is the beginning of the IQ sampling operation but only one antenna of the receiver (anchor node) carries out the sampling operation. Only one single IQ sample is performed per 1 μs, totaling eight IQ samples. Afterward, a series of sampling and switching operations begin, which is referred to switch-sample period. The switch and sample slot last 1 μs or 2 μs; in this research we consider 1 μs time slot. For every switch slot, the RF switch device switches from one antenna to another, so that another antenna can acquire a single IQ sample during the next sample slot. Since this research considers a 1 μs time slot, there are 74 sample slots in the switch-sample period. Figure 4 depicts all operations during the CTE.

Bluetooth low energy signals can be considered narrowband when using 1 MHz bandwidth in indoor scenarios where the typical delay spread is between 20 ns and 60 ns [28]. Therefore, BLE satisfies the narrowband premise in Section 3, if we take into account all the cited assumptions as well, the mathematical model is the same as Equations (Equation 2) and (Equation 3) in addition to the phase shift due to the RF switch. Without AWGN, the phase shift between two consecutive samples in the reference period was reported to be about $80^{\circ }$–$100^{\circ }$ [29]. These numbers double for two consecutive samples in the switch-sample period. As a result, it is imperative to develop a phase compensation to make the DOA method work properly. As previously mentioned, the transmitter sends a continuous signal representing digital ones which is the CTE where its carrier frequency $f_{c}$ is between 2402 MHz and 2480 MHz depending on the used channel. The narrowband incoming signal can be expressed in a complex format in Equation (Equation 27) [18],
(27)u(t)=Re(s(t)ej2πfct)=I(t)cos2πfct−Q(t)sin2πfct
where *t* is the time, and s(t)=I(t)+jQ(t) is called the complex envelope. However, that frequency in order of gigahertz is too high for the ADC, so the receiver RF front-end performs complex downconversion also known as quadrature demodulation. That operation outputs the in-phase and quadrature (IQ) components of u(t) in the baseband in such a way that the central frequency ($f_{c}$) corresponds to a DC [29]. As a result, the IQ components can be expressed in Equation (Equation 28),
(28)$$\begin{array}{cc} s(t) & =Ae^{j(2\pi \left(f_{\Delta }+f_{o}\right)t+\psi )}\\ \; & =\underset{I(t)}{\underset{︸}{A\cos(2\pi \left(f_{\Delta }+f_{o}\right)t+\psi )}}+j\underset{Q(t)}{\underset{︸}{A\sin(2\pi \left(f_{\Delta }+f_{o}\right)t+\psi )}}, \end{array}$$
where *A* is the amplitude, ψ is the initial phase, $f_{\Delta }$ = 250 kHz is the frequency deviation considering LE 1M physical layer [4], and $f_{o}$ is the carrier frequency offset (CFO) which is in order of 10 kHz [29]. Without loss of generality, let us consider that IQ samples from the reference period correspond to the first antenna of the L-shaped array. From Equation (Equation 28), the phase shift between two consecutive samples of the reference period without considering AWGN is evidenced by Equation (Equation 29),
(29)$$h_{1}\left(t+\Delta t\right)=e^{j2\pi \Delta f\Delta t}h_{1}\left(t\right),$$
where Δ t = 1 μs and $\Delta f=f_{\Delta }+f_{o}$. In other words, it is possible to estimate the phase shift over a 1 μs time period by using the samples of the reference period. However, we are interested in Δf since we can use it to estimate the phase shift over other time periods. Assuming the carrier frequency offset is constant during the CTE, we can estimate Δf by calculating the average of the phase difference between two consecutive IQ samples of the reference period, as shown in Equation (Equation 30),
(30)$$\hat{\Delta f}=\frac{1}{7(2\pi \Delta t)}\sum_{i=1}^{7}\arg(\frac{h_{1}\left(t+i\Delta t\right)}{h_{1}\left(t+\left(i-1\right)\Delta t\right)})$$

By adopting the Round Robin switch pattern, each antenna from the L-shaped array carries out IQ sampling sequentially, as shown in Figure 5. Note that the switch pattern begins in the last reference period slot. As a result, the L-shaped array samples 75 IQ samples in total, 74 samples from the switch sample period, and 1 sample from the last reference period slot. Moreover, the array observation, in this case, is defined as one single sequence of the Round Robin pattern, that is, when all antennas in the array complete the IQ sampling. Observe that antenna *k* performs an IQ sampling 2(k−1) μs after the array observation starts. It means that the phase shift is $e^{j2\pi \Delta f\Delta t_{k}}$ without considering AGWN as indicated by Equation (Equation 31),
(31)$$h_{k}\left(t+\Delta T_{k}\right)=e^{j2\pi \Delta f\Delta t_{k}}h_{k}\left(t\right),$$
where $\Delta T_{k}=2\left(k-1\right)\;\mu s$. We can generalize the observation of Equation (Equation 31). As a result, let $h_{ss}\left(t\right)$ be an array observation of a Round Robin sequence that begins at timestamp *t*, the array $h_{ss}\left(t\right)$ relates to h(t) by the phase shift matrix due to the RF switch labeled as O in accordance with Equation (Equation 32),
(32)$$h_{ss}\left(t\right)=\left[\begin{array}{c} h_{1}\left(t\right)\\ h_{2}\left(t+\Delta T_{2}\right)\\ \vdots \\ h_{2M-1}\left(t+\Delta T_{2M-1}\right) \end{array}\right]\left[\begin{array}{c} h_{1}\left(t\right)\\ h_{2}\left(t\right)e^{j2\pi \Delta f_{c}\Delta T_{2}}\\ \vdots \\ h_{2M-1}\left(t\right)e^{j2\pi \Delta f_{c}\Delta T_{2M-1}} \end{array}\right]=O\left[\begin{array}{c} h_{1}\left(t\right)\\ h_{2}\left(t\right)\\ \vdots \\ h_{2M-1}\left(t\right) \end{array}\right]=Oh\left(t\right),$$
such that,
$$\Delta T_{k}=2\left(k-1\right)T_{slot},\;1\leq k\leq 2M-1,$$
where $T_{slot}=1\mu s$ and the RF switch phase shift matrix is a diagonal matrix defined in Equation (Equation 33),
(33)$$O≜\mathrm{diag}\left(1,e^{j2\pi \Delta f_{c}\Delta T_{2}},\cdots ,e^{j2\pi \Delta f_{c}\Delta T_{2M-1}}\right)\in C^{\left(2M-1)\times (2M-1\right)}.$$

DOA methods such as MUSIC can estimate multiple DOAs during their execution, so radar applications sending sounding signals and measuring when their own signal is received from different reflections can take full advantage of that capability by identifying multiple copies of their own reflected signal. However, in IoT radio communication systems where anchors are employed to locate multiple tags, this is not possible in practice with low-cost single receiver anchor nodes operating at a single RF channel at a given time such as in Bluetooth receivers [30]. That is, if more than one BLE tag sends a signal to an anchor node at the same time and frequency resources, the signal to interference and noise ratio would be too low for that radio receiver to detect transmission reliably.

For example, a receiver could not be able to decode both transmitter IDs of the transmitters reliably as each transmission would interfere with the other. Therefore, in this scenario, DOA methods can only estimate a single DOA only per execution. As result, the L-shaped array observation model of one sequence of the Round Robin pattern is defined in Equation (Equation 34),
(34)$$h_{ss}\left(t\right)=Oh\left(t\right)=Oa\left(\alpha ,\beta \right)s\left(t\right)+n_{h}\left(t\right).$$

Note, s(t) is a scalar that represents one single signal in opposition to the vector s(t) found in Equations (Equation 2) and (Equation 3). Moreover, the number of array observations depends on the number of antennas, that is, $N=\left\lfloor \frac{75}{\left(2M-1\right)}\right\rfloor$. Notably, 75 is the number of IQ samples and 2M−1 is the number of antennas in the L-shaped array, such that 2M−1∈[3,75]. Observe that, if 75 is not divisible by 2M−1, the last 75mod(2M−1) IQ samples are not used.

MUSIC was devised considering that all antennas perform IQ sampling at the same time, which is not the case for Bluetooth specification. Thus, without a phase compensation, the accuracy of Unitary R-D Root MUSIC is totally compromised, as shown experimentally in Section 7. The DOA method receives *N* array observations as the input shown in Equation (Equation 35),
(35)$$H_{ss}=\left[\begin{array}{cccc} h_{ss}\left(t_{1}\right) & h_{ss}\left(t_{2}\right) & \cdots & h_{ss}\left(t_{N}\right) \end{array}\right].$$

From Equation (Equation 32), we know that $O^{-1}h_{ss}\left(t\right)=h\left(t\right)$. Thus, the DOA method needs to apply the RF switch compensation matrix ($O^{-1}$) into the N array observation matrix ($H_{ss}$) as expressed in Equation (Equation 36),
(36)$$H=O^{-1}H_{ss},$$
where $H=\left[\begin{array}{ccc} h(t_{1}) & \cdots & h(t_{N}) \end{array}\right].$ Note that in the real world, the equality in Equation (Equation 36) is an approximation, since the RF switch compensation matrix ($O^{-1}$) takes an estimation of Δf calculated in Equation (Equation 30). Moreover, the Unitary R-D Root MUSIC needs to obtain matrices X and Y, which are the *N* array observations from x-axis and y-axis ULA, respectively, as defined in step 1 Section 4. To do that, observe that X is equal to the first *M* rows of H, and Y is equal to the last *M* rows of H. As a result, step 1 needs an additional operation to obtain matrices X and Y prior to covariance calculations.

## 6. Modified Unitary R-D Root MUSIC

The implemented solution optimizes the Unitary R-D Root MUSIC by exploiting Bluetooth’s radio communication system design where only a single BLE tag transmits a signal at a time, as discussed in Section 5. It means that the signal subspaces, $U_{S,x}$ and $U_{S,y}$, is a column vector. As a result, the implemented solution can void applying the time-consuming EVD and instead apply the Power Method, which is a much simpler algorithm. Experimentally, we found the Power Method converge mostly in four iterations only in our solution. Moreover, the computation of the RF switch compensation matrix is performed in a linear time complexity instead of executing the inverse of a matrix with a cubic time complexity. The implemented solution utilizes a finding-root method based on EVD that does not require computing complex arithmetic despite the polynomial having complex coefficients and roots. However, the implemented solution employs the ideal array response. The real array response must be empirically found and plugged into the implemented solution. Refer to [18,31,32] to know how to compute the real array response.

The objective of the optimization is to reduce the memory consumption and execution time of Unitary R-D Root MUSIC to attain satisfactory portability to run in constrained embedded systems. Thus, the algorithms were implemented from scratch in the C99 programming language, except the inverse of sine, the inverse of a tangent, and squared root, which are functions from *math.h*. The tailor-made numerical methods include the Power Method, an EVD, which is an adaptation of [33] that consists of the Shifted QR Algorithm, the Balancing technique, Hessenberg decomposition, and auxiliary linear algebra algorithms such as the norm of a vector and multiplication of a matrix with a vector. Since one of the objectives of the implemented solution is to attain a minimal memory footprint as much as possible, it does not use the *complex.h* library from C programming language. Instead, it has a data structure for complex numbers with two variables representing the real and imaginary parts and functions for complex multiplication, addition, division, and conjugation. Notably, the implemented solution only employs *math.h* and *stdint.h* libraries, reassuring its minimal computational resources consumption goal and portability.

Due to the switching protocol of Bluetooth 5.1, more operations are required in step 1 of Section 4. That is, the method collects samples from the reference period and *N* array observations ($H_{ss}$) by performing the Round Robin switch pattern. Subsequently, the implemented solution calculates $\hat{\Delta f}$ from Equation (Equation 30) using the samples from the reference period. Afterward, it applies the RF switch phase compensation (Equation (Equation 36)) to estimate the matrix H. Then, it separates the IQ samples of the x-axis ULA from the y-axis one. More specifically, X is composed of the first *M* rows of the estimated H, while Y is the last *M* rows. Finally, it calculates the two covariance matrices, $R_{xx}$ and $R_{yy}$ from Equations (Equation 19) and (Equation 20).

The first and simpler optimization concerns Equation (Equation 19). As discussed in [34], the matrix X is big for constrained embedded devices since it contains many IQ samples that are complex numbers. In fact, if the implemented solution uses all the IQ samples, X will occupy 600 bytes considering the single-precision floating point. By performing the sample covariance matrix, the code may have to store a temporary matrix $X^{H}$ as well, which would double the memory consumption. The implemented solution does not store $X^{H}$. To analyze how it is possible, let us define V as a matrix that stores $X^{H}$. Normally, the standard way to multiply two matrices, particularly ${\hat{R}}_{xx}=XV$, is evidenced by Equation (Equation 37),
(37)$${\hat{r}}_{xx}\left(i,j\right)=\left(\frac{1}{N}\right)\sum_{k=1}^{N}x\left(i,k\right)v\left(k,j\right),\forall i,j=1,\cdots ,M.$$

Since $V=X^{H}$, then $v\left(k,j\right)=\bar{x}\left(j,k\right)$; therefore, Equation (Equation 37) could be written as indicated by Equation (Equation 38),
(38)$${\hat{r}}_{xx}\left(i,j\right)=\left(\frac{1}{N}\right)\sum_{k=1}^{N}x\left(i,k\right)\bar{x}\left(j,k\right),\forall i,j=1,\cdots ,M.$$

Moreover, since ${\hat{R}}_{xx}$ is Hermitian, which means ${\hat{r}}_{xx}\left(j,i\right)=\bar{{\hat{r}}_{xx}}\left(i,j\right)$, thus the solution applies matrix multiplication only on its upper triangular part; therefore, Equation (Equation 38) is remodeled as the set of relations in (Equation 39),
(39)$$\begin{array}{cc} {\hat{r}}_{xx}\left(i,j\right) & =\left(\frac{1}{N}\right)\sum_{k=1}^{N}x\left(i,k\right)*\bar{x}\left(j,k\right),\\ {\hat{r}}_{xx}\left(j,i\right) & =\bar{{\hat{r}}_{xx}}\left(i,j\right),\\ \forall i,j & =i,\cdots ,M. \end{array}$$

From Equation (Equation 39), the implemented solution estimates the covariance matrix using the same matrix X twice by applying an element-wise conjugate transpose operation, thus it does not need to store $X^{H}$. Furthermore, it only computes the upper triangular part of ${\hat{R}}_{xx}$. To sum up, that approach saves execution time by half and memory usage in the order of MN. That is an important improvement, since calculating the sample covariance matrix is the second most time-consuming operation, as shown in Section 7. The same optimization is carried out in Equation (Equation 20) for the y-axis ULA.

Another minor optimization concerns the computation of the RF switch compensation matrix, which requires the inverse matrix calculation of O defined in Equation (Equation 33). The Gauss–Jordan elimination is a popular algorithm to calculate the inverse of a matrix whose complex is $O(n^{3})$ [35]. However, since O is a diagonal matrix in which the generic form of its elements is known, we can avoid applying the time-consuming inverse matrix calculation. From complex arithmetic, we know that $\left(e^{j\theta }\right)e^{-j\theta }=1$, thus, the inverse of O is in line with Equation (Equation 40),
(40)$$O^{-1}=\mathrm{diag}\left(1,e^{-j2\pi \Delta f_{c}\Delta T_{2}},\cdots ,e^{-j2\pi \Delta f_{c}\Delta T_{2M-1}}\right)\in C^{\left(2M-1)\times (2M-1\right)}.$$

Since $e^{-j2\pi \Delta f_{c}\Delta T_{k}}=e^{-j2\pi \Delta f_{c}\left(k-1\right)\Delta T_{2}}$, then $e^{-j2\pi \Delta f_{c}\Delta T_{k}}={\left(e^{-j2\pi \Delta f_{c}\Delta T_{2}}\right)}^{k-1}$ for 2≤k≤2M−1. By defining, $z=e^{-j2\pi \Delta f_{c}\Delta T_{2}}$, the inverse of O can be redefined in a more compact form as evidenced by Equation (Equation 41),
(41)$$O^{-1}=\mathrm{diag}\left(z^{0},z^{1},\cdots ,z^{2M-2}\right)\in C^{\left(2M-1)\times (2M-1\right)}.$$

From the redefinition in Equation (Equation 41), we can derive the Relation (Equation 42),
(42)$$O^{-1}\left(k,k\right)=O^{-1}\left(k-1,k-1\right)z,\;2\leq k\leq 2M-1.$$

Thus, to calculate $O^{-1}$, the implemented solution only needs to set $O^{-1}\left(1,1\right)=1$, compute $z=e^{-j2\pi \Delta f_{c}\Delta T_{2}}$, and apply Equation (Equation 42) successively for k=2⋯2M−1 to take advantage of the previous computation. As a result, the implemented solution does not need to compute each element of $O^{-1}$ explicitly, that is $e^{-j2\pi \Delta f_{c}\Delta T_{k}}$, which may require computing the finite Maclaurin series 2M−2 times to calculate all of 2M−2 elements, except the first, which is 1. Notably, the finite Maclaurin series is a well-known method to evaluate complex numbers by computers, as illustrated in Equation (Equation 43),
(43)$$e^{jx}=\underset{\cos(x)}{\underset{︸}{\left(1-\frac{x^{2}}{2!}+\frac{x^{4}}{4!}-\cdots \frac{{\left(-1\right)}^{n}}{(2n)!}x^{2n}\right)}}+j\underset{\sin(x)}{\underset{︸}{\left(x-\frac{x^{3}}{3!}+\frac{x^{5}}{5!}-\cdots \frac{{\left(-1\right)}^{n}}{(2n+1)!}x^{2n+1}\right)}},$$
where *n* is the number of elements. Moreover, to reduce memory consumption, the implemented solution just needs to store the diagonal of $O^{-1}$ as a column vector and compute the element-wise (Hadamard product) of that vector with $h_{ss}\left(t\right)$ defined in Equation (Equation 32). More specifically, let $d={\left[\begin{array}{cccc} z^{0} & z^{1} & \cdots & z^{2M-2} \end{array}\right]}^{T}$ be the cited column vector, since $h_{ss}\left(t\right)\circ d=O^{-1}h_{ss}\left(t\right),$ the implemented solution calculates H by applying the Hadamard product of d in each element of $H_{ss}$ as shown in Equation (Equation 44),
(44)$$h\left(t_{i}\right)=h_{ss}\left(t_{i}\right)\circ d\;i=1,2,\cdots ,N,$$
replacing Equation (Equation 36). Particularly, the computation in Equation (Equation 44) is faster than (Equation 36), since the RF switch compensation is a matrix in Equation (Equation 36), whereas in Equation (Equation 44) it is a vector d. Moreover, Algorithm 1 calculates the RF switch compensation ($O^{-1}$) whose complexity is O(n).
**Algorithm 1:** computation of the RF switch compensation ($O^{-1}$)
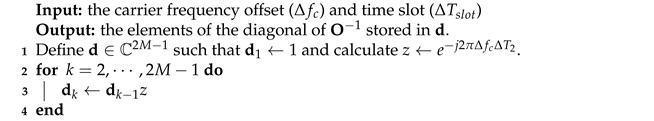


We can compute the two covariance noise subspaces ($U_{N,x}$ and $U_{N,y}$) indirectly via their respective signal subspace ($U_{S,x}$ and $U_{S,y}$) by Equations (Equation 45) and (Equation 46) [36],
(45)$$U_{N,x}{\left(U_{N,x}\right)}^{H}=I_{n}-U_{S,x}{\left(U_{S,x}\right)}^{H},$$
(46)$$U_{N,y}{\left(U_{N,y}\right)}^{H}=I_{n}-U_{S,y}{\left(U_{S,y}\right)}^{H}.$$

Remember that the implemented solution only estimates one single DOA, which means, d=1. Thus, the signal subspace is composed of only one eigenvector, which means we can apply the Power Method [37] that only estimates one eigenvector, which is the signal subspace, as proved in the next paragraph. The noise subspace is composed of N−1 eigenvectors corresponding to the smallest eigenvalues. Therefore, if we calculate the covariance noise subspace directly, we would apply an EVD method that computes all eigenvectors and eigenvalues. In addition, computing all of them requires a very complicated and time-consuming algorithm in addition to more memory footprint.

Notably, the complexity of the QR Algorithm, a typical method for EVD, is $6n^{3}+O\left(n^{2}\right)$ per iteration [38], not to mention the Hessenberg decomposition that could be performed before the QR Algorithm, and an algorithm to create the noise subspace from the probable unsorted pairs of eigenvalue-eigenvector. However, since they could be unsorted, to construct a noise subspace a reasonable approach seems to apply a sorting algorithm that could have an average complexity between O(nlog(n)) to $O(n^{2})$ [39]. While the Power Method has a complexity of $O(n^{2})$ per iteration, it calculates the signal subspace, requires very simple computations, and experimentally we found it converges mostly in four iterations only in our solution. Figure 6 depicts the algorithm overview of the covariance noise subspace computation. The left one computes the covariance directly while its fastest version (the right one) calculates the covariance indirectly via signal subspace.

To guarantee the Power Method will converge, the matrix must be diagonalizable, there must exist only one eigenvalue with the greatest absolute value, and it must be a real number [40]. For example, considering $\lambda_{i}\in R,i=1,\cdots ,M$ to be eigenvalues of a diagonalizable matrix, if  $|\lambda_{1}\left|>\right|\lambda_{2}\left|\geq \cdots \geq \right|\lambda_{M}|$ then the cited matrix satisfies the convergence requirements. The matrix $C_{x}$ is a real covariance matrix, thus it is symmetric [41]; therefore, it is diagonalizable, and its all eigenvalues are real numbers [42].

The Power Method outputs the eigenvector and its associated eigenvalue, which is the greatest in magnitude. Here, we prove that the greatest eigenvalue in magnitude is the one of the signal subspace. Note that $C_{x}$ is a real covariance matrix, hence, it is positive semi-definite [41], which means all eigenvalues are non-negative. The line-of-sight (LOS) component of the received signal that constitutes the eigenvalue of the signal subspace is greater than the eigenvalues of the noise subspace [24], and since they are all non-negative, it is not possible to have eigenvalues of the noise subspace equal to or greater than the one of signal subspace in magnitude. Therefore, the eigenvalue of the signal subspace is the greatest in magnitude, hence its corresponding eigenvector is the signal subspace. The same analysis goes to $C_{y}$.

The implemented Power Method (Algorithm 2) does not compute the eigenvalue since the solution only needs the eigenvector. We considered K=30 and $tol=10^{-6}$. We carried out thousands of experiments and we verified that in most cases the Algorithm 2 takes 4 to 5 iterations to converge, and 30 iterations are much more than enough in all experimental instances, hence for k>30 we assume the algorithm fails to compute the signal subspace.

Finding all roots of a polynomial is a difficult computational task that requires time-consuming methods, and to aggravate the problem, the polynomial in question has complex coefficients with complex roots. Classical algorithms that operate directly on the polynomial function, such as the Newton–Raphson method, Secant method, and Brent’s method, may be hard to work in practice. They only estimate one single root, some are guaranteed to converge if only certain conditions are satisfied and might not work on complex-valued polynomials, and the initial point must be chosen wisely since it could impact their convergence [43,44]. Although they can be extended to estimate multiple roots, they are highly sensitive to computing error since they operate directly on the polynomial function. That is the reason practical computer eigenvalue solvers, such as in MATLAB [45], hardly resemble these root-finding algorithms [46]. Instead, they apply EVD on the polynomial’s companion matrix to find the roots. Since such a matrix is non-symmetric, the implemented solution estimates the roots of polynomial $p_{x}\left(z\right)$ (or $p_{y}\left(z\right)$) defined in Equation (Equation 22) (or (Equation 23)) by applying the Shifted QR Algorithm in which its implementation is an adaptation of the algorithm found in [33]. The Shifted QR Algorithm is a well-known method that performs exceptionally well in practice and works even on non-symmetric matrices. It is an EVD method and an improved version of the standard QR Algorithm by including the shifting technique for rapid convergence. This algorithm calculates the eigenvalues of the companion matrix of $p_{x}\left(z\right)$ (or $p_{y}\left(z\right)$), which are its roots. To simplify, let us consider a polynomial of the form p(z), where it could be either $p_{x}\left(z\right)$ or $p_{y}\left(z\right)$.
**Algorithm 2:** Power Method
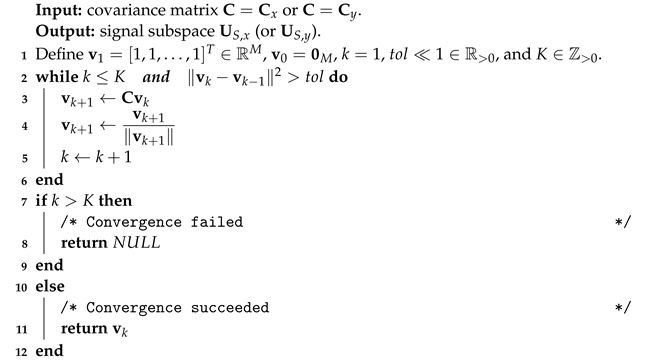


The companion matrix of the polynomial p(z) is defined in Equation (Equation 47) [43],
(47)$$P≜\left[\begin{array}{ccccc} 0 & 1 & 0 & \ldots & 0\\ 0 & 0 & 1 & \ldots & 0\\ 0 & \ddots & \ddots & \ddots & \vdots \\ \vdots & \vdots & \ddots & 0 & 1\\ -c_{0} & -c_{1} & \ldots & -c_{2M-4} & -c_{2M-3} \end{array}\right]\in C^{\left(2M-2)\times (2M-2\right)},$$
where $c_{i}=a_{i}/a_{2M-2}\forall i=0,1,\cdots ,2M-2$. That is, the companion matrix by definition relates to a polynomial in which its highest degree coefficient is one ($a_{2M-2}=1$), that is the reason we have to divide all coefficients by $a_{2M-2}$ as indicated by Equation (Equation 48),
(48)$$\frac{p(z)}{a_{2M-2}}=\frac{a_{0}}{a_{2M-2}}+\frac{a_{1}}{a_{2M-2}}z+\cdots +\frac{a_{2M-2}}{a_{2M-2}}z^{2M-2}=c_{0}+c_{1}z+\cdots +z^{2M-2}.$$

The matrix P is in a complex domain, but the implemented solution executes the Shifted QR Algorithm for real-valued matrices only. To circumvent this problem, that algorithm solves the complex EVD via equivalent real formulation by converting the complex-valued companion matrix into a real-valued one. That is, the (2M−2)×(2M−2) complex eigenvalue problem in Equation (Equation 49),
(49)(Re(P)+jIm(P))·(u+jv)=λ(u+jv)
can be reformulated as (4M−4)×(4M−4) real matrix problem in accordance with Equation (Equation 50) [47],
(50)Re(P)−Im(P)Im(P)Re(P)︸PR·uv=λuv.

However, the eigenvalue (λ) could still be a complex number since the matrix in problem (Equation 50) is non-symmetric ([33], pp. 486–487). That would apparently require complex arithmetic, but the implemented Shifted QR Algorithm circumvents it. Refer to [33] for more detail. The eigenvalue decomposition of $P_{R}$ in the reformulated problem (Equation (Equation 50)) outputs two times the number of eigenvalues of P and the complex-valued ones come in pairs of $\left(\lambda ,\bar{\lambda }\right)$ since $P_{R}$ is a real matrix [46,48]. The implemented solution can easily detect which one, λ or $\bar{\lambda }$, is the eigenvalue of P by verifying which is the root of the polynomial p(z). Moreover, the EVD of $P_{R}$ will increase the computations since its size is two times greater than P, but since the number of antennas (*M*) is small for an ULA in IoT devices, we can afford this small overburden, especially because the elimination of complex arithmetic could partly or even totally compensate this computational increment.

However, before executing the Shifted QR Algorithm the implemented solution applies the balancing technique, and afterward, it reduces the matrix to Hessenberg form. Both algorithms are an adaptation of [33] for the implemented solution. The balancing technique is a method to reduce the rounding error sensitivity of eigenvalues during the execution of EVD. Hessenberg decomposition transforms a matrix into a simpler one (Hessenberg form) to speed up the execution time of the Shifted QR Algorithms.

Algorithm 3 outputs 4M−4 eigenvalues in which 2M−2 are the roots of the polynomial p(z), and the other 2M−2 are not roots but the complex conjugate of the cited eigenvalues, as explained previously. Since d=1, ideally the implemented solution needs to find only one single root closest to the unit circle and inside it. However, due to AWGN, that root does not need to be inside the unit circle, in mathematical terms,
(51)$$\begin{array}{cc} arg min & \;{\left(\left|\lambda_{i}\right|-1\right)}^{2}\\ s.t. & \;p(\lambda_{i})=0\\ \; & \;\lambda_{i}\in \left\{\lambda_{1},\lambda_{2},\cdots ,\lambda_{4M-4}\right\}. \end{array}$$

**Algorithm 3:** Polynomial Finding Roots Method
   **Input**: The polynomial p(z) in which $p\left(z\right)=p_{x}\left(z\right)$ or $p\left(z\right)=p_{y}\left(z\right)$.
   **Output**: The roots of p(z) and their conjugate, $\lambda_{1},\lambda_{2},\cdots ,\lambda_{4M-4}$.
**1** Define the companion matrix $P\in C^{\left(2M-2)\times (2M-2\right)}$ of p(z) as described in Equation (Equation 47).
**2** Solve the complex EVD via equivalent real formulation as explained previously. Therefore, let’s define a real matrix from P, that is,
PR≜Re(P)−Im(P)Im(P)Re(P)∈R(4M−4)×(4M−4).
**3** Apply the Balancing technique in $P_{R}$ to reduce the rounding errors sensitivity of eigenvalues.
**4** Reduce the balanced $P_{R}$ to Hessenberg form to speed up the execution of Shifted QR Algorithm.
**5** Apply the Shifted QR Algorithm to the Hessenberg form of the balanced $P_{R}$ to get its eigenvalues, $\lambda_{1},\lambda_{2},\cdots ,\lambda_{4M-4}$.


The implemented solution applies Algorithm 4 to solve the optimization problem (Equation 51). However, instead of applying the complex absolute operator (|·|), the algorithm uses its squared value (${\left|\cdot \right|}^{2}$), to avoid calling the square root method in every iteration of line three. The square root method usually is an iterative algorithm and it needs to converge to a point. However, modern processors have a built-in circuit for square roots. Despite that, by avoiding that operation the execution time of Algorithm 4 is slightly decreased. In lines 2–6, observe that the algorithm finds the eigenvalue ($\lambda_{solution}$), which is the closest to the unit circle. However, its conjugate also is the closest to the unit circle since ${\left({\left|{\bar{\lambda }}_{solution}\right|}^{2}-1\right)}^{2}={\left({\left|\lambda_{solution}\right|}^{2}-1\right)}^{2}$. Thus, in lines 7–9, the algorithm finds which one is the root of the polynomial p(z). That is, if the eigenvalue $\lambda_{solution}$ is also a root, then ${\left|p\left(\lambda_{solution}\right)\right|}^{2}\ll 1$ which means ${\left|p\left(\lambda_{solution}\right)\right|}^{2}<{\left|p\left({\bar{\lambda }}_{solution}\right)\right|}^{2}$, otherwise, its conjugate is the root. In that way, the algorithm does not need to check if the eigenvalue is the polynomial root in every iteration (lines 2–6) which saves execution time.
**Algorithm 4:** Find the eigenvalue which is the root closest to the unit circle
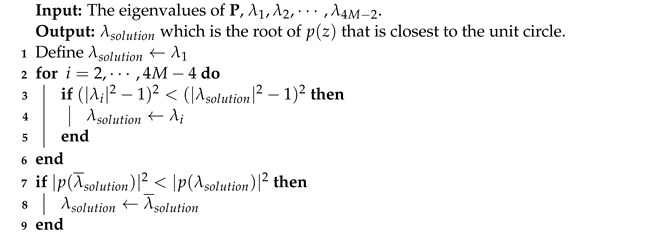


## 7. Experiments

The objective of the experiment consisted of showing the modified Unitary R-D Root MUSIC (implemented solution) works and is feasible for commercial embedded IoT devices. The experiment comprises two parts. In the first part, to check the effectiveness of the RF switch compensation, we compared the implemented solution with the cited compensation and without it in a MATLAB environment only. The second part is more complex. It is composed of a simulation in MATLAB and the real world. In summary, the baseband signals were artificially generated in MATLAB to be the input of the implemented solution developed in C99 programming language and executed in a constrained embedded IoT device. Therefore, we could measure the memory footprint, execution time, energy consumption, and accuracy. To be as accurate as possible, such a device executed the implemented solution only without any operating systems or software layer, that is, the implemented solution was bare-metal programmed. An overview of the experiment is depicted in Figure 7.

### 7.1. Experimental Setup

For both parts of the experiments, the artificial baseband signals were generated using the 5G Toolbox, Phased Array System Toolbox, and Communication Toolbox provided by MATLAB. The simulation parameters are shown in Table 1. The Tapped Delay Line TDL-E channel model (corresponding to Line of Sight propagation) was employed to simulate the multipath propagation phenomenon in indoor environments alongside Additive White Gaussian Noise (AWGN). The center carrier frequency and the frequency deviation correspond to the CTE, and the simulation also randomly generated the CFO between [−30 kHz,+30 kHz] using Gaussian distribution. The CFO values were chosen based on the empirical experiment in [49], which estimated that 99% of CFO values in Bluetooth were within such interval. The L-shaped array is composed of seven isotropic antennas, that is, four antennas for each ULA. This number of antennas is small, thus, it is reasonable for IoT devices. The distance between antennas is half of the Bluetooth signal wavelength.

To measure the memory footprint (RAM and Flash), execution time, energy consumption, and accuracy, we employed a PCA10056 development kit that comes with an nRF52840 System-on-Chip (SoC) having an Arm Cortex-M4 of 64 MHz with a Floating-Point Unit (FPU). The SoC did not use an operating system or software layers. The hardware floating-point instructions and hardware floating-point linkage (*-mfloat-abi=hard*) were activated so the processor could fully operate the FPU. All devices of the nRF52 series have support for BLE, and although nRF52840 does not have Bluetooth Direction Finding capability, it is almost identical to other nRF52 and nRF53 devices that do have it. Notably, the nRF52 and nRF53 devices are a well-known series of constrained IoT devices developed by Nordic Semiconductor with a radio module of Bluetooth 5.1 or later versions and come with an Arm Cortex-M4 or Arm Cortex-M33 processor. Another popular one is the EFR32BG22 SoC series developed by Silicon Labs, which has the direction-finding capability, ARM Cortex-M33 with 76.8 MHz, up to 512 kB of flash, and 32 kB of RAM. Table 2 shows some SoCs with Bluetooth Direction Finding capability.

The implemented solution used the single-precision floating-point (FP32), under IEEE 754-2008 specification, since the FPU of Arm Cortex-M4 does have support for FP32 only. Hence, floating-point operations with FP32 attain the fastest execution time as empirically shown in [34], and achieve the same accuracy as the other two floating-point formats. The half-precision floating-point (FP16) is another format employed by ARM processors; however, that format is used as a storage format only for Arm Cortex-M4. When operating in FP16, the processor promotes FP16 into FP32 before and demotes it after every calculation [50]. Those operations create a small overhead that could increase the Flash consumption and the execution time. A slower execution time may translate into more energy consumption. Another supported format is double-precision floating-point (FP64), but the FPU of Arm Cortex-M4 does not have support for FP64 at all. Thus, the C compiler emulates FP64 calculations [51,52], and that emulation creates an excess of computations culminating in a substantial increment of execution time and Flash usage. In fact, the execution time of DOA methods using FP64 was shown to be about 20 times slower than ones using FP32 [34].

To measure the energy consumption, we connected the Otti Arc (power measurement tool) to the nRF52840, and to measure the execution time, we utilized the Saleae logic analyzer. However, in both measurements we needed to activate a General-Purpose Input/Output (GPIO) port, thus a GPIO was set high and low before and after the execution of the algorithm. So, we could check when the method started and finished to properly carry out the two measurements. Thus, energy usage is slightly overestimated.

Additionally, we measured the stack memory consumption and the relative execution time of the principal operations in the implemented solution. There is no dynamic memory usage. We define the relative execution time as the running time of an operation divided by the execution time of the implemented method in percentage.

Moreover, we calculated the Root Mean Squared Error (RMSE) of accuracy considering a 500 azimuth-elevation pair for each SNR. In mathematical terms, the RMSE of accuracy is defined in Equation (Equation 52),
(52)$$RMSE=\sqrt{\left(\frac{1}{L}\right)\sum_{i=1}^{L}\left({\left(\theta_{i}-{\hat{\theta }}_{i}\right)}^{2}+{\left(\phi_{i}-\hat{\phi_{i}}\right)}^{2}\right)},$$
where L=500 is the number of estimated azimuth-elevation pair, and θ and $\hat{\theta }$ is the actual azimuth and its estimation in degrees, respectively. Similarly for the elevation (ϕ). The SNR values were 5 dB, 10 dB, 15 dB, 20 dB, 25 dB, and 30 dB. In total, we analyzed 6000 different pairs.

### 7.2. Results and Discussions

Figure 8 shows the comparison between the implemented solution with RF switch compensation (a) and the one without it (b). Both were implemented in MATLAB. The implemented solution without the RF switch compensation is totally inaccurate, whereas the one with it attains much better accuracy as the SNR increases which demonstrates the effectiveness of such compensation. As previously explained, this experiment was the only one carried out totally in a simulation environment. For the next experiments, the implemented solution was run in an nRF52840 SoC.

Accuracy is the most important performance criterion of a DOA method. Some companies [53,54] reported an average accuracy of about 5 degrees with an average position accuracy of around 1 m. Thus, it would be desirable for the implemented solution to achieve such a value. Figure 9 shows the RMSE of the accuracy for each SNR run in an nrf52840 SoC, which have almost the same values as one in MATLAB. We clearly see that as the noise decreases the accuracy improves. However, low accuracy is observed between SNR 5–10 dB. However, it should not be a concern since the minimum SNR value for Bluetooth to operate reliably is between 10–15 dB. With less than 10 dB, it is most likely that the receiver fails to decode and the cyclic redundancy check fails as well [55]. With SNR slightly higher than 10 dB, the RMSE values attain less than 5 degrees, reaching our desired result.

Nevertheless, the experiments did not consider non-idealities of the antenna array and RF front-end; in fact, the antennas have an ideal isotropic characteristic. As mentioned previously, the implemented solution considers an ideal array response. Since those imperfections deteriorate the accuracy, one should compute a real array response and use it in the implemented solution. The real array response should attenuate the problem, but not completely solve it.

Table 3 shows each main operation of the implemented solution with its respective stack memory consumption and relative execution time. All operations have a stack memory consumption of a few hundred bytes; thus, setting maximum stack memory to be 512 kB in the microcontroller is enough since none of the stack memory surpasses that value. The fifth operation concerns the finding-root method, which took up an incredible 83.17% of the method’s execution time. It is not difficult to verify that no other operation demands so much computation as the fifth one, since finding polynomial roots composed of complex roots and coefficients is a difficult numerical computing task. Notably, Algorithm 3 tackles this problem by applying the Shifted QR Algorithm, a complicated method that requires two pre-processing methods to speed up its convergence and accuracy. On the other side of the spectrum, computing the companion matrix (four operations) is the least demanding task. It computes the polynomial coefficients defined in Equation (Equation 25), and afterward, it constructs the real companion matrix, which is $P_{R}$ defined in Equation (Equation 50).

Moreover, the second operation is the second most time-consuming operation. It involves computing Equation (Equation 39) and the conversion of a complex-valued into a real-valued covariance as explained in step 2 Section 4. Notably, even though the implemented solution applies the optimization (Equation Equation 39), which reduces the computation by half and stack memory by 600 B, that operation comes in second, which shows the importance of that optimization in reducing both the execution time and stack memory. Furthermore, the third operation constitutes the Power Method (Algorithm 2), and Equations (Equation 45) and (Equation 46). As explained in Section 6, the optimization avoids the execution of an EVD method, which could be as computationally demanding as the fifth operation. Finally, the first operation is composed of Algorithm 1 and Equations (Equation 44) and (Equation 30), which are computationally inexpensive.

Table 4 shows the memory footprint, execution time, and energy consumption. From these values, the implemented solution satisfies the memory requirements for an nRF52, nRF53, and EFR32BG22 SoC series, as can be verified in [56,57,58]. The execution time is 16.2 ms. By comparison, the standard MUSIC implemented in [59] takes about 18 ms to 133 ms with a median of roughly 31 ms to estimate a single DOA of one ULA in the same SoC (nRF52840). That means that even though the implemented solution estimates DOAs from two ULAs, it is still almost two times faster than the median of the standard MUSIC. A 2D standard MUSIC was implemented in C programming language based on parallel computing using a Digital Signal Processor of 1 GHz and an L-shaped antenna [22]; however, despite the parallelization and a much more powerful processor, it takes about 190 ms. Nonetheless, the implemented solution could be much slower than the modified 2D Unitary TLS ESPRIT in [34]. Its 1D version was developed in the same SoC, and it takes about 0.855 ms using the same precision format (single-precision floating-point). Hence, we can roughly estimate that its 2D version for L-shaped arrays could be two times that value, that is, 1.71 ms if it were developed.

Coin batteries are used for small electronic devices [60], including constrained IoT ones. We found that the capacity of such batteries ranges from 1 mAh to 2000 mAh [61] in a well-established global distributor of semiconductors and electronic components. That means, considering the implemented solution as the only source of energy consumption, the nRF52 series can run from 16,574 to more than 33 million times approximately. Therefore, the implemented solution can be used for battery-powered small embedded devices. However, it is worth mentioning the experiment did not measure the RF front-end, since we did not employ a real array of antennas. Therefore, in practice, we considered that the measurement was evaluated after IQ sampling.

## 8. Conclusions

This paper presented a novel Unitary R-D Root MUSIC for L-shaped arrays tailor-made for constrained embedded systems using a switching protocol defined by Bluetooth, and a more insightful implementation perspective that is usually not addressed sufficiently in papers. More precisely, the implemented solution exploits the radio communication system design to speed up its execution, that is, it applies a simple Power Method instead of the time-consuming EVD. It also has a root-finding method that circumvents complex arithmetic despite being used for complex polynomials.

In theory, all antennas in the array sample the signal at each antenna port at the same time; however, Bluetooth specifies that the array has an RF switch, so each antenna samples the signal at a different time. Therefore, the theoretical model was slightly modified to consider the RF switch. We showed that without an RF switch phase compensation, the accuracy of the implemented solution was totally compromised. Therefore, we developed a method of RF switch phase compensation and conceived a linear complexity algorithm to compute the phase compensation matrix.

To prove the solution viability, we carried out experiments on energy consumption, memory footprint, accuracy, and execution time in a commercial constrained embedded IoT device (nRF52080) without operating systems and software layers. Notably, except for accuracy, other performance criterion usually are not carried out in research; however, in ours, they were too important to be neglected. With such measurements, we showed the implemented solution viability for IoT devices verified by us, its few milliseconds execution time, and its good accuracy achievement.

## Figures and Tables

**Figure 1 sensors-23-03356-f001:**
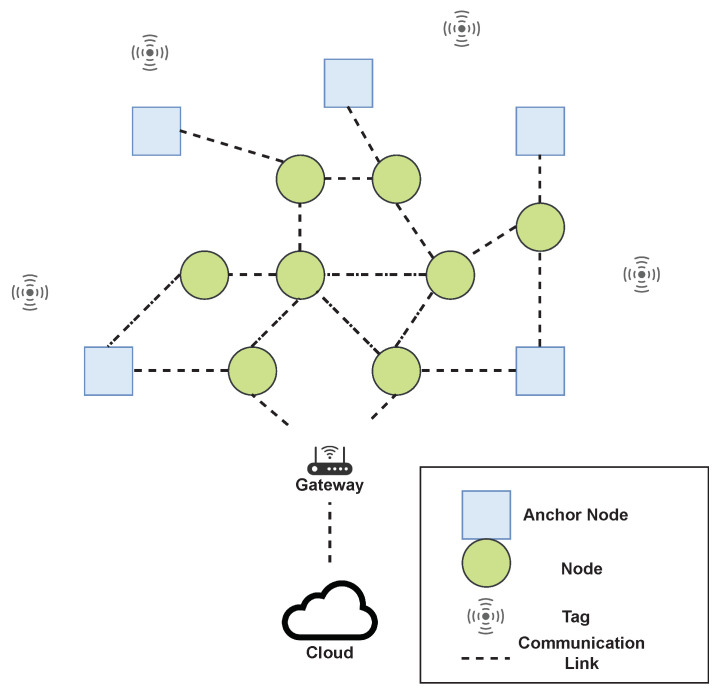
An IoT mesh network with a DOA-based positioning system.

**Figure 2 sensors-23-03356-f002:**
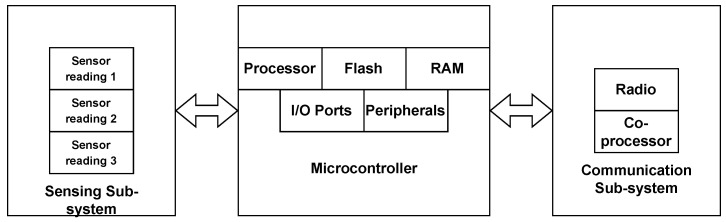
An overview example of a node’s hardware architecture.

**Figure 3 sensors-23-03356-f003:**
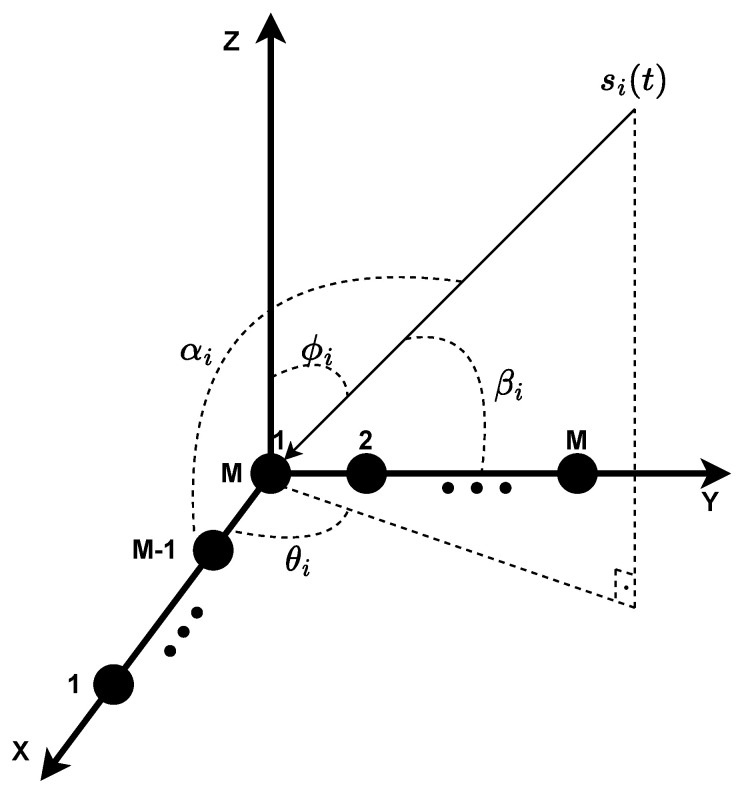
Depiction of L-shaped array with its antennas (black dots), angles, and the signal direction.

**Figure 4 sensors-23-03356-f004:**
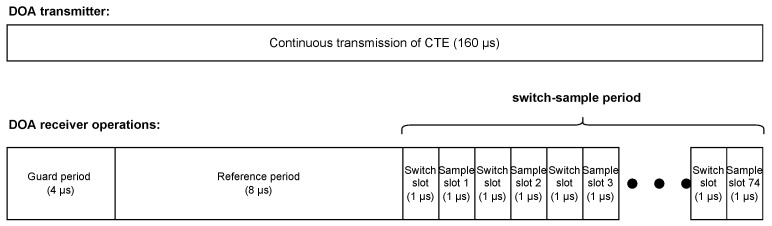
Depiction of the transmitter and receiver operations.

**Figure 5 sensors-23-03356-f005:**
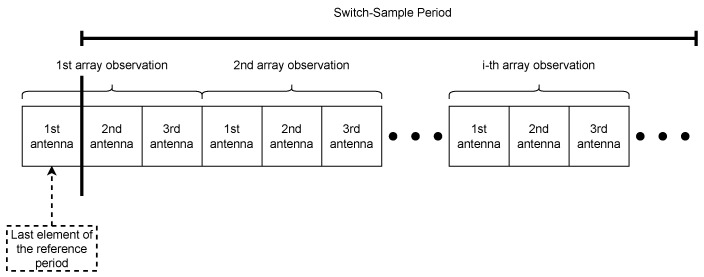
Example of the Round Robin switch pattern of a L-shaped array with three antennas. Only the sample slots are shown.

**Figure 6 sensors-23-03356-f006:**
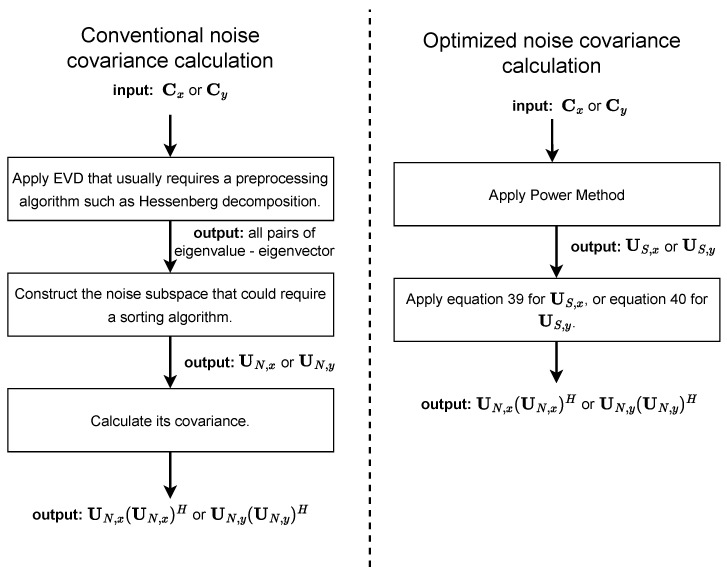
Algorithm overview of the two covariance noise subspace computations. The left method calculates that covariance directly, but the right one calculates indirectly via the signal subspace.

**Figure 7 sensors-23-03356-f007:**
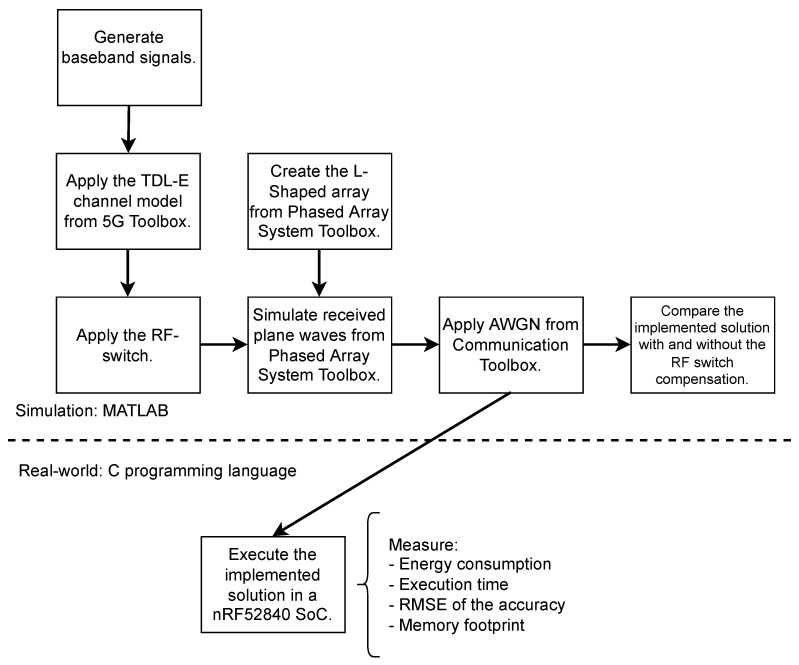
Overview of the experiment.

**Figure 8 sensors-23-03356-f008:**
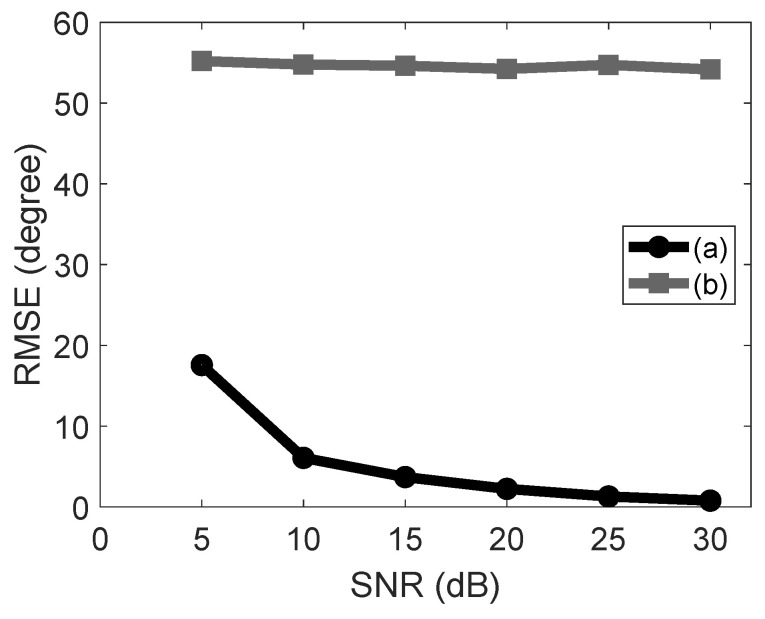
The implemented solution with RF switch compensation (a) and without it (b) run in a simulation (MATLAB).

**Figure 9 sensors-23-03356-f009:**
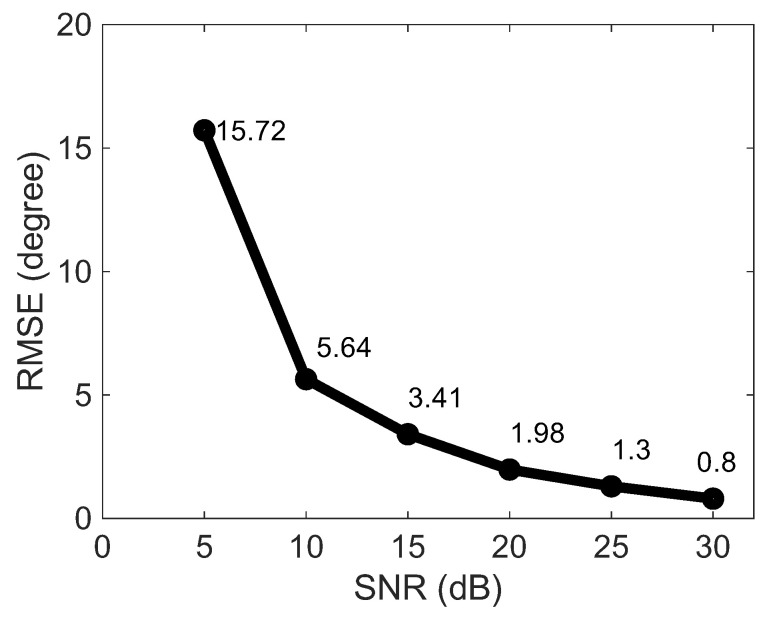
RMSE of the accuracy over the SNR run in a nRF52840 SoC.

**Table 1 sensors-23-03356-t001:** Simulation parameters in MATLAB.

Simulation Parameters	
Antenna array type	L-shaped array
Antenna type and frequency ranges	Isotropic of 2 GHz–3 GHz
Distance between antennas (Δ)	λ/2
Frequency deviation ($f_{\Delta }$)	250 kHz
Carrier frequency offset ($f_{o}$)	[−30 kHz, +30 kHz]
Center carrier frequency ($f_{c}$)	2.4 GHz
Sampling frequency	1 MHz
Channel model	TDL-E + AWGN
Number of antennas	7

**Table 2 sensors-23-03356-t002:** SoCs with Bluetooth Direction Finding capability.

SoC	Processor	Flash Memory	RAM	Does It Have FPU?
nRF52833	ARM Cortex M4 64 MHz	512 KB	128 KB	Yes
nRF52811	ARM Cortex M4 64 MHz	192 KB	24 KB	No
nRF52820	ARM Cortex M4 64 MHz	256 KB	32 KB	No
nRF5340	ARM Cortex-M33 128/64 MHz	1 MB	512 KB	Yes
EFR32BG24 Series	ARM Cortex M33 78 MHz	Up to 1536 KB	Up to 256 KB	Yes

**Table 3 sensors-23-03356-t003:** Stack memory consumption and relative execution time of principal operations.

Operation	Total Stack Memory	Relative Execution Time
Estimate the frequency and apply the RF phase compensation	128 B	2.94%
Calculate the covariance matrices	112 B	7.67%
Compute the noise covariance matrix	440 B	5.27%
Compute the companion matrix	64 B	0.93%
Find the polynomial roots	224 B	83.17%

**Table 4 sensors-23-03356-t004:** Measurement values of the implemented solution executed in nRF52840 SoC.

RAM Consumption	Flash Memory Consumption	Execution Time	Energy Consumption
4.72 KB	20.33 KB	16.2 ms	181 nWh

## Data Availability

Some data and software are available upon request from (T.T.).

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
