# Peer review of "Direction of Arrival Method for L-Shaped Array with RF Switch: An Embedded Implementation Perspective"

_sensors, 2023, doi:10.3390/s23063356_

Round 1

Reviewer 1 Report

A few mentioned clarifications are required.

In introduvtion part , 

the author needs to focus on recent research gaps in DoA for shaped array, such as how it is compatible with 5G applications and what are the drawbacks.

How does your proposed mathematical model perform identification if more than one single IQ  sample is performed per 1us. What is its impact on round-robin switching and sample period?

Figure.4 shows the optimized noise covariance calculation algorithm overview. Equations 39 and equation 40 are useful to rectifiy the noise subspace in two demensional DoA estimations with L shaped array.

Explain how algorithm.4  is useful to analyse the multiple antenna spectrum sensing in cognitive radio networks.

In figure .6 The comparison graph between RF switch comparison and without it, should be differentiated as (a) and (b) and label has been included in a graph instead of described in text part. In x-axis please mentioned units of measurement.

In section 6.1 experimental setup, How have you generated artificial baseband signal from 5G toolbox.. if possible, could you please explain what is the initial setup required for the artificial baseband signal in 5G toolbox?

In figure.5 you have clearly mentioned that the c-program is used for experiment. Can you explain How can you incorporate c-program in IoT devices, ARM-Cortex-M4 and ARM Cortex -M33.

Author Response

Dear Sir/Madam, 

Thank you for your comments.
I submitted the revised paper. The modifications are highlighted in blue and a changes.pdf file is attached mentioning the minor changes.

My replies to your comments:

"the author needs to focus on recent research gaps in DoA for shaped array, such as how it is compatible with 5G applications and what are the drawbacks."
I'm afraid it is a misunderstanding, the paper does not focus on the 5G radio interface, but here DOA is a generic method for any radio interface. This paper focused on the recent Bluetooth Direction Finding feature, the challenge of implementing a complicated numerical method (DOA method) in constrained embedded systems with an L-shaped antenna using an RF switch specified by Bluetooth Direction Finding, and its solution is a development of an existing DOA method optimized for such application.

"How does your proposed mathematical model perform identification if more than one single IQ  sample is performed per 1us. What is its impact on round-robin switching and sample period?"

As explained in the paper, the Bluetooth Direction Finding specification defines that only one single IQ sample is performed per sampling slot. Oversampling would reduce the quantization noise but the control of ADC is not available in the chipset, thus it is impractical, nevertheless, the impact would be minimal as the quantization noise is not dominating in the reception performance. The round-robin switch pattern is a fast and simple method to sample all antennas and is supported by the BLE standard.

"Figure.4 shows the optimized noise covariance calculation algorithm overview. Equations 39 and equation 40 are useful to rectifiy the noise subspace in two demensional DoA estimations with L shaped array."

Yes, you understand this correctly.

"Explain how algorithm.4  is useful to analyse the multiple antenna spectrum sensing in cognitive radio networks."

Multiple antenna spectrum sensing was not part of our study, it focused on the implementation of DOA in IoT constrained devices. Developing multiple access schemes based on this multiple antenna spectrum sensing requires separate studies.

"In figure .6 The comparison graph between RF switch comparison and without it, should be differentiated as (a) and (b) and label has been included in a graph instead of described in text part. In x-axis please mentioned units of measurement."

We have fixed it.

"In figure.5 you have clearly mentioned that the c-program is used for experiment. Can you explain How can you incorporate c-program in IoT devices, ARM-Cortex-M4 and ARM Cortex -M33."

The process of uploading firmware into an embedded system is a routine task that requires an integrated development environment (IDE) with a cross-compiler for the target device. However, since this is a commonly known practice, we have chosen not to include it in this paper.

Reviewer 2 Report

In this paper, a novel Unitary R-D Root MUSIC was proposed. Moreover, experiments verified the effectiveness of the proposed method.  Generally, this paper is well written and can be accepted after revisions. My comments are listed in the following.

1) In experiments, comparisons with more recent algorithms are necessary.

2) For DOA estimation, some robust algorithms for smart antenna system should be discussed (e.g. adaptive filtering methods, LCMV, MVDR…).

Author Response

Dear Sir/Madam,

Thank you for your comments. I submitted the revised paper.

The modifications are highlighted in blue and a changes.txt file is attached mentioning the minor changes.

My reply to your comments:

This paper presents the validation of an optimized DOA method designed for embedded systems that employ an L-shaped array with an RF switch, as defined by Bluetooth Direction Finding specifications. The implemented solution was developed from scratch using C99 programming language, which is the de facto language for firmware implementation in embedded systems. Firmware development is much more complicated than in MATLAB, since it requires a lower-level programming language (C99) without any pre-made numerical methods. While there are robust DOA methods available, they are not directly comparable in their standard form, as they are not designed for this type of application. To make such comparisons, they would need to be implemented from scratch in the firmware, which would require a separate study for each of them.

Reviewer 3 Report

This manuscript presents a novel Unitary R-D Root MUSIC for L-shaped arrays tailor-made for such devices using a switching protocol defined by Bluetooth, and a more insightful implementation perspective. The authors have a deep understanding of the research field and a clear description of the method. Although the manuscript has several mistakes inside, it is sound in all.

Author Response

Dear Sir/Madam,

Thank you for your comments. I submitted the revised paper. The modifications are highlighted in blue and a changes.txt file is attached mentioning the minor changes.

All the several mistakes were fixed.
